# Expert Multinational Consensus Statement for Total Intravenous Anaesthesia (TIVA) Using the Delphi Method [note 1]

**DOI:** 10.3390/jcm11123486

**Published:** 2022-06-17

**Authors:** Giulia Uitenbosch, Daniel Sng, Hugo N. Carvalho, Juan P. Cata, Hans D. De Boer, Gabor Erdoes, Luc Heytens, Fernande Jane Lois, Anne-Françoise Rousseau, Paolo Pelosi, Patrice Forget, David Nesvadba

**Affiliations:** 1School of Medicine, Medical Sciences and Nutrition, University of Aberdeen, Aberdeen AB25 2ZD, UK; daniel.sng@nhs.scot; 2Anesthesiology and Perioperative Medicine, Universitair Ziekenhuis Brussel, Vrije Universiteit Brussel, 1090 Brussels, Belgium; carvalho.hn@gmail.com; 3Department of Anesthesiology and Perioperative Medicine, Division of Anesthesiology, Critical Care, and Pain Medicine, University of Texas MD Anderson Cancer Center, Houston, TX 77030, USA; jcata@mdanderson.org; 4Department of Anesthesiology, Pain Medicine and Procedural Sedation and Analgesia, Martini General Hospital, 9728 NT Groningen, The Netherlands; hd.de.boer@mzh.nl; 5University Department of Anaesthesiology and Pain Medicine, Inselspital, University Hospital Bern, University of Bern, 3010 Bern, Switzerland; gabor.erdoes@insel.ch; 6Departments of Anesthesiology and Neurology, University Hospital Antwerp, 2650 Edegem, Belgium; luc.heytens@gza.be; 7Malignant Hyperthermia Research Unit, Born Bunge Institute, University of Antwerp, 2000 Antwerpen, Belgium; 8Department of Anaesthesia and Intensive Care Medicine, CHU Liege, Domaine du Sart-Tilman, 4000 Liege, Belgium; fernande.lois@chuliege.be; 9Burn Center and Intensive Care Department, University Hospital of Liege, University of Liege, 4000 Liege, Belgium; afrousseau@chuliege.be; 10Anaesthesia and Critical Care, San Martino Policlinico Hospital, IRCCS for Oncology and Neurosciences, 16132 Genoa, Italy; ppelosi@hotmail.com; 11Department of Surgical Sciences and Integrated Diagnostics (DISC), University of Genoa, 16132 Genoa, Italy; 12Department of Anaesthesia, NHS Grampian, Aberdeen AB25 2ZD, UK; patrice.forget@abdn.ac.uk (P.F.); david.nesvadba@abdn.ac.uk (D.N.); 13Epidemiology Group, Institute of Applied Health Sciences, School of Medicine, Medical Sciences and Nutrition, University of Aberdeen, Aberdeen AB25 2ZD, UK

**Keywords:** TIVA, total intravenous anaesthesia, volatile anaesthesia, anaesthetic techniques, peri-operative anaesthesia

## Abstract

**Introduction:** The use of total intravenous anaesthesia (TIVA) has been well established as an anaesthetic technique over the last few decades. Significant variation in practice exists however, and volatile agents are still commonly used. This study aims to determine the motivations and barriers for using TIVA over the use of volatile agents by analysing the opinion of several international anaesthetists with specific expertise or interests. **Methods and participants:** The Delphi method was used to gain the opinions of expert panellists with a range of anaesthetic subspecialty expertise. Twenty-nine panellists were invited to complete three survey rounds containing statements regarding the use of TIVA. Anonymised data were captured through the software REDCap and analysed for consensus and prioritisation across statements. Starting with 12 statements, strong consensus was defined as ≥75% agreement. Stability was assessed between rounds. **Results:** Strong consensus was achieved for four statements regarding considerations for the use of TIVA. These statements addressed whether TIVA is useful in paediatric anaesthesia, the importance of TIVA in reducing the incidence of postoperative nausea and vomiting, its positive impact on the environment and effect on patient physiology, such as airway and haemodynamic control. **Conclusions:** Using the Delphi method, this international consensus showed that cost, lack of familiarity or training and the risk of delayed emergence are not considered obstacles to TIVA use. It appears, instead, that the primary motivations for its adoption are the impact of TIVA on patient experience, especially in paediatrics, and the benefit to the overall procedure outcome. The effect of TIVA on postoperative nausea and vomiting and patient physiology, as well as improving its availability in paediatrics were considered as priorities. We also identified areas where the debate remains open, generating new research questions on geographical variation and the potential impact of local availability of monitoring equipment.

## 1. Introduction

The intravenous anaesthetic propofol has been widely used to induce general anaesthesia since its introduction in the 1980s. However, what has divided clinical opinion across the board is the use of intravenous agents for both induction and maintenance of anaesthesia, in a technique called propofol-based total intravenous anaesthesia (TIVA) [1]. The literature suggests several potential benefits for the use of TIVA over volatile agents. It is thought to be more environmentally friendly as it reduces the production of waste anaesthetic gases which is attributed to volatile anaesthesia [2,3]. TIVA may also be associated with a positive effect on patient physiology, such as more stable haemodynamic conditions due to high dose opiate analgesia, less reliance on airway to achieve hypnosis, as well as a reduction in the incidence of post-operative nausea and vomiting [1,4,5]. Several studies also suggest that TIVA may improve the overall survival in cancer patients [6,7,8]. However, arguments against the use of TIVA may include a higher incidence of awareness and delayed emergence from anaesthesia, especially in paediatric cases [9,10]. Nevertheless, popularity of TIVA seems to be increasing [11,12,13]. Additionally, TIVA being a newer technique than volatile anaesthesia, it may not be as widely popular, possibly leading to geographic variations in familiarity or training. Lastly, until recently, intravenous agents were generally thought to be more expensive than older, widely used volatile anaesthetics [14]. It could be argued, however, that by inherently considering volatile anaesthesia a cheaper technique, volatile agents would be administered more liberally, increasing the cost overall. As a result of these varying opinions on the adoption of TIVA in the anaesthetic arsenal, it is no surprise that significant variations in practice exist regarding anaesthesia administration.

As outlined above, there is a wealth of information available regarding the potential advantages and disadvantages of the use of TIVA over volatile agents. Most of these data, however, do not differentiate any variations in practice from a subspecialty, geographical or academic perspective. We formulated the hypothesis that by collecting expert opinions from a variety of anaesthetic subspecialties across the world, we may be able to better understand the motivations for the use of TIVA and whether global challenges or barriers exist, which may result in lower popularity in its use.

## 2. Methods

### 2.1. Model

The Delphi method is a well-known process used for obtaining group consensus in healthcare, as it encourages decision making and new ideas to be formed by whole group feedback [15,16]. The method involves several rounds of anonymous questionnaires, where after each round structured feedback and aggregated responses from previous rounds are presented to panellists (Figure 1).

### 2.2. Panellist Recruitment

In order to obtain perspectives on anaesthetic techniques from a broad range of professional and geographical settings, an international panel of anaesthetists was recruited with a range of subspecialty interests. All communication was carried out via email. We created an email template outlining the project and explaining the requirements of prospective panellists, and contacted a large number of specialists across the world, only including individuals to the panellist list once they replied citing interest in their involvement. Whilst assured that responses would be anonymous, participants were invited to be listed as collaborators on a future publication, as an additional incentive for participation. Once prospective panellists agreed to participate in the study, their email addresses were added to the final participant list for survey distribution. In terms of the selection of individuals to invite, we selected panellists by either identifying a key opinion leader in an anaesthetic field (e.g., neuro-anaesthesia, obstetric anaesthesia, pain management) or by reaching out to committees of international anaesthetic societies, including the European Association of Cardiothoracic Anaesthesiology and Intensive Care (EACTAIC), European Society of Anaesthesiology and Intensive Care (ESAIC), UK Society for Intravenous Anaesthesia (SIVA) and European Society for Regional Anaesthesia and Pain Therapy (ESRA), requesting for an interested member to volunteer as panellist. Clinicians were contacted from across all continents; however, the majority of specialists who responded and agreed to participate in the project were European (79%), creating potential bias in terms of the geographical popularity of TIVA use.

Anonymous demographic data were gathered from the panel during Round 1 to determine location of practice and experience. This showed that twenty-one (72.4%) panellists practice in tertiary care centres, whereas eight (27.6%) are based in secondary care. Furthermore, the average amount of anaesthetic practice was twenty years, with two panellists possessing less than ten years of experience, thirteen possessing between ten and twenty years of experience and fourteen possessing between twenty and fifty years of experience. We hoped that this extensive collective knowledge would reveal interesting opinions from the panel.

### 2.3. Statements

A list of twelve statements was formulated regarding the use, advantages and disadvantages of general use of TIVA as an anaesthetic technique. In order to formulate these statements, the team conducted a review of the available scientific literature describing the use of TIVA in a range of anaesthetic settings, whilst also attempting to collect evidence of its possible benefits and disadvantages to its use. The search identified factors such as practicality, environmental impact, physiological benefits, the availability of training and knowledge of the technique and the usefulness of TIVA in subspecialties such as paediatric anaesthesia and onco-anaesthesia (Appendix B). Once identified through the literature search, these aspects of TIVA use were used as a basis on which to build the statements for Round 1 of the survey.

For the consecutive survey rounds, statements were modified according to comments offered by the panellists, facilitated through the addition of an open text box at the end of each round. This was arranged to provide panellists with an opportunity to add suggestions or opinions which could be agreed on to form consensus in the wider group.

### 2.4. Survey Design and Data Collection

Study data were collected and managed using REDCap electronic data capture tools hosted at the University of Aberdeen [17,18]. REDCap (Research Electronic Data Capture) is a secure, web-based application designed to support data capture for research studies, providing: (1) an intuitive interface for validated data entry; (2) audit trails for tracking data manipulation and export procedures; (3) automated export procedures for seamless data downloads to common statistical packages; and (4) procedures for importing data from external sources [19].

REDCap was also used for building the surveys themselves. Three surveys were created: Round 1 proposed the initial 12 statements, each linked to a 5-level Likert scale of agreement, ranging from “Not at all” to “Very much”. A free text box was included at the end of the first survey, for panellists to add comments or suggestions on their experience with using TIVA.

Round 2 contained the same statements as the first round, with the possibility to be edited based on comments and additional statements made by panellists during round one. Furthermore, during round two, each participant would be presented with their answer to each statement during the previous round, as well as the aggregated responses and percentage of agreement of other panellists to the statements.

Round 3, the final survey, saw the removal of statements which had not reached a strong positive or negative consensus over the course of the two previous rounds: participants would be asked to rank the remaining statements, which had indeed reached strong consensus, in terms of prioritisation or importance.

### 2.5. Data Analysis

Through the exportation of the results from REDCap, we analysed the overall consensus for each statement by gathering the number of times participants had “agreed” or “strongly agreed” to a statement, or inversely whether a majority “somewhat disagreed” or “strongly disagreed”. The threshold for a strong consensus was established as ≥75%, and the stability of each answer was determined as a change in agreement proportions of <10% between each round.

Prioritisation of statements in the final round was calculated as the proportion of participants who had agreed on each statement to be a priority.

### 2.6. Ethical Review

All participants in this study were contacted directly via email and were only included in the participant list once a clear affirmative response was received to the invitation to participate.

As per the Medical Research Council and the NHS Health Research Authority, a submission was made to determine whether this study required NHS Research Ethics Committee (REC) review. A formal confirmation was obtained attesting that NHS REC review would not be required for this project.

## 3. Results

### 3.1. Round 1

Twenty-nine out of thirty-one participants who originally agreed to take part in the project completed the first survey round. Two panellists were unable to participate due to local holidays. A breakdown of the panellists’ countries of provenience is shown in Figure 2, and a full list of panellists’ anaesthetic fields of interest is shown in Appendix B. Panellists stated their level of agreement with all twelve statements, which resulted in three statements reaching strong consensus (Appendix B). Eighteen participants added additional statements in the free text box regarding further challenges to the use of TIVA. These comments primarily raised issues around the availability of TIVA equipment such as pumps and monitoring, the practicality of the use of TIVA in terms of equipment and type and length of surgery, as well as patient factors such as needle phobia (although rare) and a history or family history of malignant hyperthermia.

### 3.2. Round 2

All twenty-nine panellists who completed Round 1 also completed Round 2. The second survey round saw the modification of four statements to incorporate the suggestions that participants had provided during Round 1 (Appendix B). The altered statements were proposed to the panellists with the inclusion of their answers from the previous round and the aggregated responses of the other panellists. At the end of the second survey, the three statements which had reached consensus after Round 1 remained stable (less than 10% change in answers between the two rounds), and a fourth reached consensus following the modification of the statement (Appendix B). An additional statement was identified with the potential to reach consensus with further modifications suggested by some of the panellists via email.

### 3.3. Round 3

Twenty-eight out of twenty-nine panellists completed the third and final round. This survey contained four statements with strong, stable consensus which participants were asked to rank in terms of priority or importance (Figure 3). It also included one modified statement for a final assessment of whether consensus could be reached (Appendix B). The modified statement received a total of 74.1% agreement therefore not reaching strong consensus, so its consequent prioritisation question was disregarded in the results. As shown in Figure 3, panellists considered post-operative nausea and vomiting and the availability of TIVA for paediatric anaesthesia as priorities. Next came the effects of TIVA on intraoperative physiology, and the consideration of the environmental impact of TIVA was prioritised last.

## 4. Discussion

The results from this Delphi survey show that despite scientific evidence and considerations regarding cost, familiarity and training, departmental preferences and even the benefits for the environment, the decision to use TIVA is largely attributed to what is best for the patient, both in terms of personal experience and physiological benefit.

It is interesting to note that participants did not collectively agree on the majority of statements put to them, and only three found a strong consensus from Round 1. This suggests that, at least sometimes, there may be a gap between the efficacy of an anaesthetic technique and its appropriateness, and that this appropriateness may depend on various reasons.

Most participants agreed that the reduced environmental impact of TIVA was important to them (Appendix A). There was also an immediate strong consensus on the fact that TIVA is a useful technique in paediatric anaesthesia, and that one of its main benefits lies in the reduced occurrence of nausea and vomiting postoperatively. This suggested early on that patient experience plays a role in the decision to use TIVA. However, only half the panellists considered the potential benefit of TIVA on cancer biology to be a significant factor in deciding its use, which may be a result of subspecialty variation or a lack of practical value.

Through feedback received in Round 1, it became clear that panellists considered practicality an important challenge for the use of TIVA: panellists were asked about practicalities (or impracticalities) as a whole in order to garner opinions on a range of practical considerations. As a result, some panellists voiced concerns over the amount of additional work involved when using TIVA if the procedure time is short, and the relative inconvenience of having to change syringes. It was also suggested that the use of TIVA may be impacted by lack of ready availability of drugs, equipment and technology such as target controlled infusion (TCI) pumps and depth of anaesthesia monitoring equipment such as processed encephalograms (pEEG), as well as patient factors such as strong needle phobia or difficult access (e.g., small children). These factors were therefore specified in the later statement in order to assess the overall opinion on whether these factors impact TIVA use. Remarkably, when these considerations were included in the statement to reflect panellists’ opinions, a strong consensus was still not reached, with the proportion of panellist agreement increasing from 34.4% to only 64.3% between rounds (Appendix B). It is worth noting at this point that when considering the practical aspect of TIVA use, despite comments suggesting that monitoring equipment availability may be a barrier to TIVA use, the anonymity of the survey renders it impossible to comment on the distribution among panellists of pEEG and TCI pump use when employing TIVA as a technique. We are, therefore, unable to establish whether a possible variation in the use of monitoring equipment among panellists may have caused interference when attempting to reach a consensus regarding TIVA practicality, or indeed whether it may be the cause of that statement not reaching consensus.

Similarly, when panellists were asked whether they believe there is strong evidence to suggest TIVA reduces the incidence of emergence agitation, initially, only 62.1% to 68.9% of panellists agreed. When feedback was added to the statement specifying that this reduction in emergence agitation may be more widely observed in paediatric anaesthesia, consensus only reached 74.1%, therefore, also not reaching a strong consensus. This suggests that the challenges and considerations proposed by panellists are observed by a few individuals, but do not necessarily reflect the experience of the majority; these differences could again be due to the variation in subspecialty expertise, geographical differences, or even as a result of working in a regional rather than a national hospital. Although the results of this survey suggest hypotheses rather identified factors, they highlight the added value of such a Delphi project, generating new research questions on variations in anaesthetic technique use. Future studies, informed by the current work, may aim to clarify the importance of local or regional aspects to be addressed when, or before, implementing TIVA.

In contrast to the abovementioned statements, which failed to reach consensus despite comments suggested by the panellists, the inclusion of feedback that patient factors, as well as the effect of TIVA on patient physiology such as haemodynamic stability and the uncoupling of airway and hypnosis, determine whether panellists use this technique, did increase the consensus between rounds from 55.1% to 75.8% (Appendix B). This is an important finding, as similarly to the consensus regarding paediatric anaesthesia and the reduction of post-operative nausea and vomiting, it is evident that patient factors constitute a substantial consideration in the decision to use TIVA regardless of subspecialty or location. While this may not be surprising as individual patient factors can be a driving factor for variation in many clinical areas, we consider it an important finding as it demonstrates objective evidence that this may specifically be the case for TIVA use as well.

There was a consistent lack of collective agreement on statements suggesting that TIVA may not be used due to financial burden, lack of training, familiarity or departmental preferences to use volatile agents. This may suggest that, regardless of subspecialty or geographical location, the main considerations for the use of TIVA are patient dependent rather than departmental or organisational. However, this may also reflect the fact that, for instance, financial inequities or local constraints lead to different limitations in practice and therefore a lack of consensus. Future surveys could identify which specific patient factors may carry more weight when deciding to use TIVA, and investigate local variations in these patient factors considering different patient demographics depending on location and anaesthetic subspecialty. Further work could also be directed at possible reasons for lack of consensus and geographical variability when considering local factors, such as financial, training and departmental aspects.

When it came to prioritising statements, it was clear that the reduction of nausea and vomiting and availability of TIVA in paediatric anaesthesia are considered to be the main priorities when deciding on its use. This was closely followed by the opinion that TIVA’s effect on the patients’ physiology should be prioritised, further demonstrating the patient focused approach to TIVA use. Interestingly, TIVA’s positive environmental impact, which in Round 1 gained a very strong agreement, was considered as a priority by the fewest participants. This suggests once more that even when moral causes such as protecting the environment are concerned, patient experience and factors play a more significant role in deciding whether to use TIVA.

### 4.1. Strengths

We used the Delphi approach to gather opinions on motivations or challenges for the use of TIVA as an anaesthetic technique, and analyse responses and feedback from experts in different anaesthetic subspecialties and geographical areas. This method was useful for this study as, due to our wish to involve an international panel, remote survey completion seemed the most practical method to collect data.

The study itself provided useful insight into the motivations that drive clinicians from various anaesthetic subspecialties to use TIVA as a technique, providing clear results on what individual panellists believe is important. Specifically, it allowed us to identify that despite location, range of anaesthetic experience and subspecialty, patient factors and overall patient experience (especially in paediatric cases) are among the main considerations when choosing which technique to use for induction of anaesthesia. We also believe the results of this survey have posed interesting questions for future research into the variability of anaesthetic practice.

### 4.2. Limitations

Due to the study design, it is, however, impossible to determine if opinions are correlated to geographical location, anaesthetic subspecialty or even the type of hospital panellists work in. This is particularly highlighted as the majority of the panellists who took part in the project are based in Europe, creating the potential for geographical bias and misrepresenting the value and use of TIVA in other continents. Further work could focus on these variations, by increasing the number of panellists from different subspecialties to highlight any specialty-specific preferences, as well as widening the geographical representation in the panel to extrapolate whether differences in opinion can be attributed to the country a panellist practices in or the subspecialty they have particular expertise in. Furthermore, this project addressed the reasons for the use of TIVA as an anaesthetic technique. Although the survey considers this in comparison with volatile agents, it does not specifically put to the panellists the benefits or otherwise of the use of the latter. In future research, a parallel survey could determine motivations for the use of volatile anaesthesia over TIVA, to establish whether there is a reciprocal relationship between the downsides of using TIVA and the reasons for using volatile agents.

## 5. Conclusions

This manuscript aimed to address the variation in practice when administering anaesthetic techniques. By asking participants to state their agreement and ranking priorities relating to the use of TIVA in their respective subspecialties, it became apparent that despite scientific evidence and considerations regarding cost, familiarity and training, departmental preferences or even the benefits for the environment, the decision to use TIVA is largely attributed to what is best for the patient, both in terms of personal experience and physiological benefit. Whilst the results of this survey may not directly impact clinical practice, we believe they provide useful insight into the motivations of TIVA use and may provide a foundation for future research on anaesthetic technique employment from an individual preference basis, departmental and equipment availability perspective and wider geographical variation. They also provide helpful opinions regarding the use of TIVA in paediatric anaesthesia, which may be looked into further to establish whether departmental changes or training curricula may be adjusted in the future to reflect this.

## Figures and Tables

**Figure 1 jcm-11-03486-f001:**
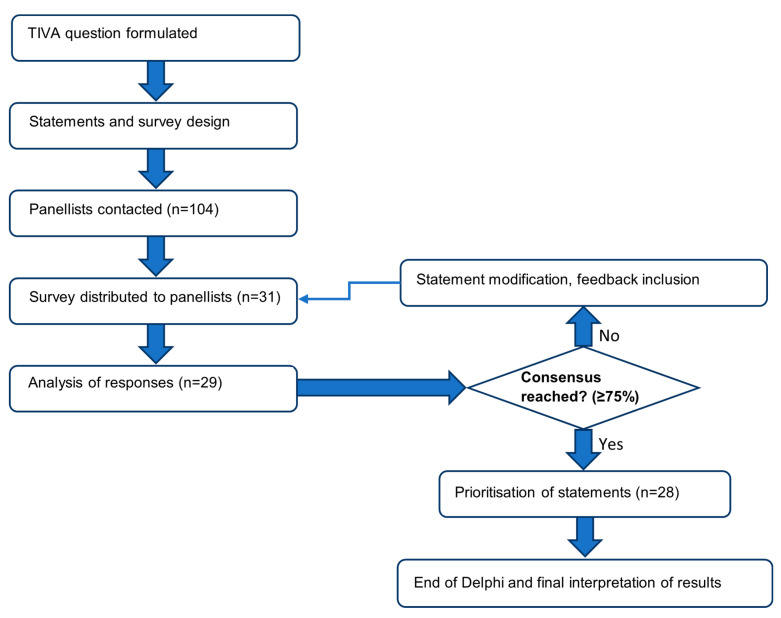
Flowchart depicting the process of the Delphi method.

**Figure 2 jcm-11-03486-f002:**
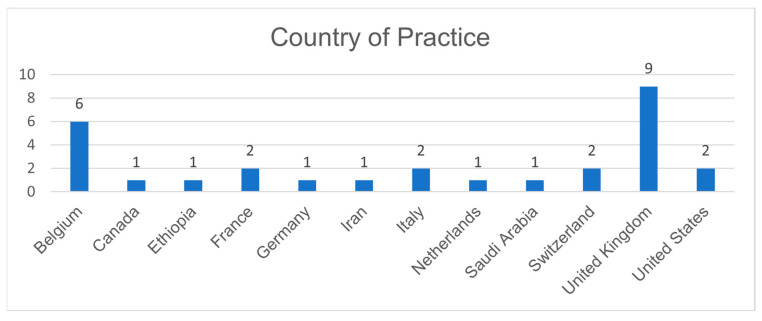
This chart shows the range of countries from which panellists were recruited.

**Figure 3 jcm-11-03486-f003:**
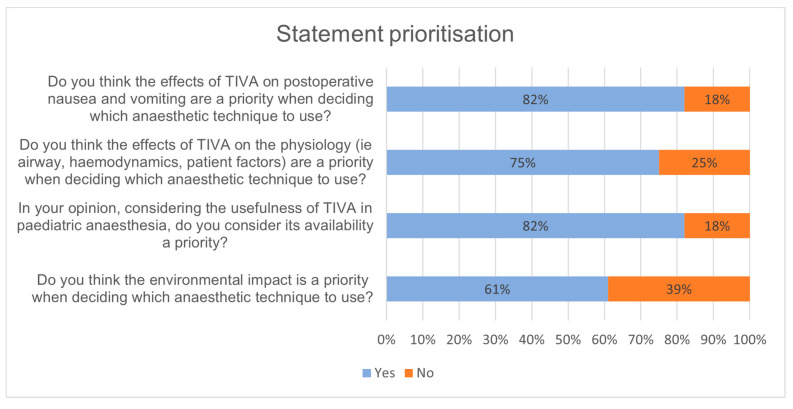
This graph shows the panellists’ prioritisation of the statements with a strong consensus.

## Data Availability

Data is contained within this article.

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
