# Peer review of "Expert Multinational Consensus Statement for Total Intravenous Anaesthesia (TIVA) Using the Delphi Method †"

_jcm, 2022, doi:10.3390/jcm11123486_

Round 1

Reviewer 1 Report

Here are some methodological issues that makes conclusions from this paper not of value to the readers.

1.To call it an international guideline is not appropriate without true representation from all countries; for example, south east asian countries are not represented.

2.The consultant group is possibly biased towards practices in Belgium (6) and United Kingdom(9).

3.There is ample literature on the environmental benefits of TIVA. Poor mix of consultants may have skewed the opinion (group thinking!).

4.Interesting to note a paper on TIVA  without mentioning processed EEG monitoring. Assumption is that TCI is prevalent, whilst it may not be correct.Do appreciate the question on awareness, though!

5.The statements of strong agreement amongst the consultants are  leading and too broad with minimal guidance for the readers. i.e: "considering the usefulness of TIVA in pediatric anesthesia...."

6.There is minimal new knowledge presented in a conclusive manner.

Author Response

Many thanks for taking the time to read this manuscript, and thank you for your helpful comments. Please see the responses below:

1.To call it an international guideline is not appropriate without true representation from all countries; for example, south east asian countries are not represented.

Answer: We appreciate that not all countries, or even all continents, are equally represented, but would like to assure you attempts were made to include as many geographical variations as possible. Unfortunately we only received responses from a select number of specialists, which is what we worked with to carry out the survey. We also agree that to call this manuscript an international guideline is not appropriate, and we want to assure you that is not the aim of the article. We have reviewed it thoroughly and have checked that it does not state (anymore) or claim to be such anywhere in the text.

2.The consultant group is possibly biased towards practices in Belgium (6) and United Kingdom(9).

Answer: As outlined above, we accept and agree that the geographical representation is somewhat limited in this case. We have included text in the manuscript to reflect this limitation and have made new suggestions in the text for future work to include a wider geographical pool of specialists. Still, we believe the opinions of the panellists who were involved are valid, and of use to the clinical community in terms of providing interesting insight into the motivation for anaesthetic technique, and we hope you agree.

3.There is ample literature on the environmental benefits of TIVA. Poor mix of consultants may have skewed the opinion (group thinking!).

Answer: As the reviewer certainly knows, a Delphi survey is a tried and tested method to extrapolate opinions and form a consensus on topics while maintaining anonymity of the panel, therefore (hopefully!) preventing group thinking (or at least providing a robust method for minimising group thinking). We agree that there is a wealth of evidence regarding the environmental benefits of TIVA, and we are not disputing that. Simply the panellists have agreed that whilst it is an important factor to consider when deciding which anaesthetic technique to use, patient safety is considered a priority over TIVA’s environmental benefit. We hope this now addresses your comment.

4.Interesting to note a paper on TIVA  without mentioning processed EEG monitoring. Assumption is that TCI is prevalent, whilst it may not be correct. Do appreciate the question on awareness, though!

Answer: We have included a paragraph in the text which addresses the fact that TCI prevalence/availability or the use of pEEG amongst the panellists is not measured in the survey. This may have interfered when attempting to come to a consensus, but as the specific practicalities added in the statement during Round 2 still did not help reach consensus, it is difficult to know whether it played a part. We hope you are happy with the addition in the text.

5.The statements of strong agreement amongst the consultants are leading and too broad with minimal guidance for the readers. i.e: "considering the usefulness of TIVA in pediatric anesthesia...."

Answer: Unfortunately we are unable to change or add to the statements regarding paediatric anaesthesia being a useful technique, as that specific wording was used by the panel to form a decision in the survey. Any mention of usefulness of TIVA in the text is made either in the context of the consensus reached by the panellist (ie the panel agreed that it was indeed useful), or as a measure of how useful it is (is it useful or not) and is not intended to sound leading. We have made changes to the wording in order to clarify this in the manuscript.

6.There is minimal new knowledge presented in a conclusive manner.

Answer: We accept there are limitations to this project, but we still believe the survey has merit, was of sound methodology and provided some interesting results into the clinical reasoning behind different anaesthetic approaches.

Reviewer 2 Report

Dear Editorial Board,

Thank you very much for allowing me to review the manuscript JCM-1751459 entitled: “Expert international consensus statement for TIVA using Delphi method”.

Summary:

In this study authors used a Delphi method to determine the motivations and barriers for using TIVA over the use of volatile agents among international anaesthetists with specific expertise or interest in TIVA. This international consensus showed that the primary motivations to use TIVA were the usefulness of TIVA in paediatric anaesthesia, the importance of TIVA in reducing the incidence of postoperative nausea and vomiting, its positive impact on the environment and effect on patient physiology such as airway and haemodynamic control.

Here are our comments below:

I appreciated the amount of work put by the authors in this study and in this manuscript preparation. The manuscript was well written in good English and was easy to read. The methodology was clearly described and was appropriate in this context. However, I have questions to ask and comments to make.

1)    Twelve statements were formulated and then distributed to the panelists. However, this important part of the methodology is poorly described. It should be more precisely explain, for instance, how the statements were chosen, who has formulated the statements…

2)    The authors have written that the panelists were worldwide dispatched. However, the panelists were mainly Europeans (23/29, 79%). This should be discussed in the discussion section. Indeed, the results are probably strongly depicting the European use of TIVA rather than a global use. Countries as China, Japan or South Korea are not represented in this study.

3)    There is a mistake in the Appendix 2, first line of the table: “Sheme 1” should be replaced by “Statement 1”.

Author Response

Thank you for your helpful comments and for taking the time to review the manuscript. Please see below the responses:

  • Twelve statements were formulated and then distributed to the panelists. However, this important part of the methodology is poorly described. It should be more precisely explain, for instance, how the statements were chosen, who has formulated the statements…

Answer: We have made some changes to the methodology section to better explain the process of our statement formulation and hope you are satisfied with the result.

  • The authors have written that the panelists were worldwide dispatched. However, the panelists were mainly Europeans (23/29, 79%). This should be discussed in the discussion section. Indeed, the results are probably strongly depicting the European use of TIVA rather than a global use. Countries as China, Japan or South Korea are not represented in this study.

Answer: We agree with you that there is limitation in the geographical variety of the panel. We would like to assure you steps were made to include clinicians from as many countries as possible, but unfortunately can only work with the ones who responded to our invitation. We have addressed this point in the text, and have included suggestions for improvement in the future.

  • There is a mistake in the Appendix 2, first line of the table: “Sheme 1” should be replaced by “Statement 1”.

Answer: Many thanks for pointing out this error, this has been corrected.

Round 2

Reviewer 1 Report

-Please replace the 'international' from the title with 'multinational'

It should read - "Expert multinational consensus statement for total intravenous anaesthesia (TIVA) using a Delphi method"

Author Response

Dear reviewer,

Many thanks the additional comment. The title has been revised as requested, with the term "international" being replaced with "multinational".